

# A multi-service data management platform for scientific oceanographic products

Alessandro D'Anca[1], Laura Conte[1], Paola Nassisi[1], Cosimo Palazzo[1], Rita Lecci[2], Sergio Cretì[2], Marco Mancini[3], Alessandra Nuzzo[1], Maria Mirto[1], Gianandrea Mannarini[2], Giovanni Coppini[2], Sandro Fiore[1], and Giovanni Aloisio[1,4]

[1]Centro Euro-Mediterraneo sui Cambiamenti Climatici (CMCC), Advanced Scientific Computing, via per Monteroni, 73100 Lecce, Italy
[2]Centro Euro-Mediterraneo sui Cambiamenti Climatici (CMCC), Ocean Predictions and Applications, via Augusto Imperatore 16, 73100 Lecce, Italy
[3]Centro Euro-Mediterraneo sui Cambiamenti Climatici (CMCC), Supercomputing Center, via per Monteroni, 73100 Lecce, Italy
[4]University of Salento, Complesso Ecotekne, via per Monteroni 73100 Lecce, Italy

*Correspondence to:* A. D'Anca (alessandro.danca@cmcc.it)

**Abstract.** An efficient, secure, and interoperable data platform solution has been developed in the TESSA project to provide fast navigation and access to the data stored in the data archive, as well as a standard-based metadata management support. The platform mainly targets scientific users and the Situational Sea Awareness high-level services such as the Decision Support Systems (DSS). These datasets are accessible through the following three main components: the Data Access Service (DAS), the Metadata Service, and the Complex Data Analysis Module (CDAM). The DAS allows access to data stored into the archive by providing interfaces for different protocols. OPeNDAP, THREDDS, and WMS are just some of the solutions that have been integrated into the TESSA infrastructure. Metadata Service is the heart of the information system of the TESSA products and completes the overall infrastructure for data and metadata management. This component enables data search & discovery, and addresses interoperability by exploiting ISO standards for geospatial data (ISO 19115 and ISO 19139). Finally, the CDAM represents the back-end of the TESSA DSS by performing on-demand complex data analysis tasks.

## 1 Introduction

TESSA (Development of **TE**chnology for **S**ituational **S**ea **A**wareness) is a research project born from the collaboration between operational oceanography research and scientific computing groups in order to strengthen the operational oceanography capabilities in Southern Italy for use by end users in the maritime sector, tourism and environmental protection. This project has been very innovative as it has provided the integration of marine and ocean forecasts and analyses with advanced technological platforms. Specifically, an efficient, secure, and interoperable data platform solution has been developed to provide fast navigation and access to the data stored in the data archive, as well as a standard-based metadata management support. This platform mainly targets scientific users and a set of high-level services such as the Decision Support Systems (DSSs) for supporting end users in managing emergency situations. In this context, the developed platform faces the lack of an efficient dissemination of



marine environmental data in order to support the Situational Sea Awareness (SSA), which is is strategically important for the maritime safety and security (Lecci et al., 2015). In fact, an updated Situation Awareness requires an advanced technological system to make data available for decision makers, improving the capacity of intervention and avoiding potential damages.

In a "data centric" perspective, in which different services, applications or users make use of the outputs of regional or global numerical models, the TESSA Data platform meets the request of near real-time access to heterogeneous data with different accuracy, resolution or degrees of aggregation. During the design phase of the platform, the results of the MyOcean project (MyO) were taken into consideration, as this project implementation was in line with the best practices of the GMES/Copernicus framework (cop). Specifically, regarding data access, MyOcean provides the scientific community with a unified interface designed to take into account various international standards (ISO 19115, ISO 19139, INSPIRE). Concerning the data management, MyOcean relies on OPeNDAP/THREDDS for tasks like map subsetting and FTP for direct download. Another valid solution is the Earth System Grid Federation (ESGF) (Cinquini et al., 2014), a federated system used as metadata service with advanced features, that will be explained later. Both of these systems support a wide range of functionalities such as data access, classification, search & discovery and download by means of a distributed architecture. The proposed architecture of the TESSA Data Platform, should offer superior and fast data management functionalities, producing datasets suitable for serving different kinds of services in near real-time.

This paper is organized as follows: in Sect.2, an architectural overview of the main Data Platform components is provided. The focus is on providing easier and unified access to the heterogeneous data produced in the frame of TESSA project. The implemented data archive and the modules that make the datasets available to the entire set of services are presented in Sect3. In Sect.4, the metadata service features are detailed from a methodological and technical point of view. The module developed in order to serve remote DSSs submission requests is described in Sect.5. Finally in Sect.6 a few use cases are presented, highlighting the operational chains which exploit the Data Platform services.

## 2   The Data Platform Architecture

As reported in Fig.1, the TESSA Data Platform is composed of three main components: the Data Access Service, the Metadata Service and the Complex Data Analysis Module.

The DAS (Data Access Service) allows the access to the Data Archive, the storage of the data produced in the frame of TESSA project. The main consumers of these services are the "intermediate users" (the scientific community) and the automatic procedures fed by produced datasets to perform the needed transformations. In this context, the Data Access Service represents the single interface to access the Data Archive of TESSA: its design has been driven by the aim to satisfy requirements such as transparency, robustness, efficiency, fault tolerance, security, data delivery, subsetting and browsing. This implies that this component must necessarily be "data-centric" and "multi-service", in order to offer access to the data, often stored in a single instance on the Data Archive, through different protocols and, therefore, different levels of functionality.

In addition, the need of interoperability with other data management platforms has driven the design toward the inclusion of well-known services:

- "domain-oriented", able to provide specific functionalities for the climate change domain;

- "general purpose", to satisfy general requirements exploiting common services and protocols.

The Metadata Service has been designed as the heart of the information system of the TESSA products, to support the activities of search & discovery and metadata management. Moreover, the Metadata Service addresses the requirements of interoperability, considered a key feature in order to provide to any potential user an easier and unified access to the different
types of data produced by the project. These requirements have been satisfied through the adoption of a metadata profile compliant to the ISO standards for geospatial data (ISO) and the implementation of an efficient Information System relying on a specific ESGF Data Node.

Finally, the CDAM supports the Situational Sea Awareness high-level services, by performing on demand complex analysis on the datasets stored in the data archive. In particular, a number of intelligent Decisions Support Systems for complex dynam-
ical environments has been designed and developed exploiting high performance computing functionalities to execute their algorithms. One of the goals is to deploy an infrastructure where a DSS requires the execution of a complex task including (i) accessing data stored in the archive and/or (ii) executing one or more preliminary tasks for processing them, (iii) storing the results in a specific section of the data archive or publishing them on other sites. Each DSS request interacts with the platform involving a three-layer infrastructure: the user client (web or mobile platform), the SSA Platform (Scalas et al., 2016) and the
CDAM. Each layer communicates with their linked tiers exchanging messages using well-known protocols and it has been developed in order to implement an high separation of concerns.

The design and the main features of these modules will be detailed in the following sections.

## 3   The Data Access Service

As stated in the previous section, the Data Access Service represents the single interface to access the Data Archive of TESSA;
specifically, it has been designed with the aim to setup a multi-service platform to access and analyze scientific data. Among the exposed services it is important to mention the THREDDS Data Server (TDS) (thr) which represents a web server that provides metadata and data access for scientific datasets, using a variety of remote data access protocols. More in detail, it offers a set of services that allow users a direct access to data produced in the context of the TESSA project namely the output of the AFS (Adriatic Forecasting System)(Guarnieri et al., 2008), (Oddo et al., 2005), (Oddo et al., 2006), AIFS (Madec, 2008),
(Oddo et al., 2009) and SANIFS (Federico et al., 2016) sub-regional models. In particular, the THREDDS includes the WMS - Web Map Service (wms, 2006), for the dynamic generation of maps starting from geographically referenced data, the direct downloading using the HTTP protocol, and the OPeNDAP (ope) service, which is a framework that aims to simplify the sharing of scientific information on the web by making available local data from remote connections, providing also variables selection and spatial and temporal subsetting capabilities. In addition, a sub-component, the Data Transformation Service (DTS), is
responsible to apply different data transformations in order to make data compliant to the different protocols offered by the DAS and therefore available to the data consumers (users or services).



## 3.1 The Dataset repository: the TESSA Data Archive

The Data Archive represents the storage for all the data produced within the TESSA project. It has been designed and implemented in order to ease the data search & discovery phase exploiting their classification. Fig.2 shows an high-level schema of the archive.

The data can be subdivided into two categories:

- historical data: permanently stored into the archive, including observations and model reanalysis and best analysis;

- rolling data: subjected to a rolling procedure, and including model worst analysis, simulations and forecasts.

Considering the different kinds of data, the data archive has been created following a tree-based topology where the directory names reflect the data classification: (e.g. historical as first level, model or observation as second level, atmospheric or

oceanographic as third level, etc.). The directory tree includes also the institute providing the data, the name of the system that produced the data and the temporal resolution of the data. This kind of classification eases and speeds up the search & discovery of data by intermediate/scientific users and automatic procedures. Concerning the "historical" archive, data stored under "obs" include observations retrieved by different instruments (satellites, in-situ buoys, vessels, etc.) and consider measurements of the main oceanographic variables (temperature, altimetry, salinity, chlorophyll concentration, etc). The spatial and the temporal

resolution and the temporal coverage are different depending on the data type.

    Data stored under "model" directory include the best data produced by the numerical models (atmospheric or oceanic) by different external providers. Typically, these data are produced assimilating observational data in order to improve the algorithm functionalities which drive the models. Spatial, temporal resolution and temporal coverage depend on the model considered.

    The data stored under "rolling" section exist no longer than 15-days, meaning that they are permanently removed from the

archive after that time period since they are superseded by better quality data, as for example updated forecasts. The "rolling" archive includes also data subjected to further processing operations saved under the proper directories: the processing could include an interpolation to a finer grid in order to increase the spatial resolution ("derived" data), a subsampling by time or by space ("generic" data) or a transformation in a new format for specific purposes ("dts" data). The access to the Data Archive is allowed only via the Data Access Service described below.

## 3.2 Implementation of the DAS-DTS modules

Fig.3 shows a high-level representation of the Data Access Service (DAS) and its strict interaction with the Data Transformation Service (DTS) and the Data Archive, as implemented in the framework of TESSA project. It is composed by cyclic workflows totally automatized that execute a set of steps on a set of data.

    This interchange system has been implemented through some high-parameterized software components which schedule and

parallelize the different jobs on an high performance computing environment. The software modules developed for this system are: the Downloader module, the DTS_organizer module and the Uploader module. Schematically, these modules download source raw data from a set of external providers and, after a set of transformations, taking into account the appropriate data



format, store them into the Data Archive and publish them through the set of services exposed. Finally the Rolling_cleaner module is used to delete obsolete files and directories in the rolling archive.

## 4 The Metadata Service and the metadata profile

The analysis of geospatial metadata standards has been the starting point for the definition of a specific metadata profile suitable for the description of the datasets produced in the frame of TESSA project. The International Standard ISO 19115 "Geographic Information-Metadata" is a standard of the International Organization for Standardization (iso, 2003a). It is a constituent of the ISO series 19100 standards for spatial metadata and defines the schema required for describing geographic information. In particular, this standard provides information about the identification, the extent, the quality, the spatial and temporal schema, spatial reference, and distribution of digital geographic data. ISO 19115 is applicable to the cataloguing of datasets and the full description of datasets, geographic datasets, dataset series, and individual features and feature properties.

The ISO 19115 standard provides approximately 400 metadata elements most of which are listed as "optional". However, it has also drawn up a "core metadata set for geographic dataset" for search, access, transfer and exchange of geographic data; metadata profiles compliant to this International Standard shall include this core to ensure homogeneity and interoperability. All metadata profiles based on ISO 19115 should include this minimum set for conformance with this international standard.

The ISO 19115 metadata standard has been widely adopted by a large number of international oceanographic and marine projects, such as the International Hydrographic Organization (iho) and SeaDataNet (sea, b). For its characteristics, it has been considered a valid solution also for TESSA project, as capable of satisfying requirements of simplicity and interoperability. Thus, a specific metadata profile has been designed for describing and cataloging TESSA products, according to the requirements of the international metadata standard ISO 19115 and its XML implementation ISO 19139 (iso, 2003b).

A detailed analysis of TESSA datasets led to the definition of the key aspects about how, when and by whom a specific set of data was collected, how the data are accessed and managed and which data formats are employed in order to obtain a general overview of data and metadata available. On the basis of this information, a minimum set of metadata has been defined able to describe the datasets produced in the frame of TESSA project. This schema includes all mandatory elements of the "core metadata set for geographic datasets" drawn up by the ISO 19115 standard for search, access, transfer and exchange of geographic data: Dataset title, Dataset Reference date, Abstract describing the dataset, Dataset language, Dataset topic category, Metadata point of contact, Metadata date stamp and On-line resource.

The ISO 19115 standard for metadata is being adopted internationally as it provides a built-in mechanism to develop a "Community profile" by extending the minimum set with additional metadata elements to suit the needs of a particular group of users.

Hence the generation of the specific TESSA Metadata Profile composed of the minimum set of metadata required to serve the full range of metadata applications (discovery and access) and other three optional elements that allow for a more extensive description and, at the same time, highlight the main features of TESSA products.



Another important step in the design of TESSA Metadata Service has been the analysis of the Information Systems most widely adopted, at international level, to support the activities of search & discovery.

In particular, the Earth System Grid Federation (ESGF) has been recognized and selected as the leading infrastructure for the management and access of large distributed data volumes for climate change research. It consists of a system of distributed and federated Data Nodes that interact dynamically through a Peer-to-Peer paradigm to handle climate science data, with multiple petabytes of disparate climate datasets from a lot of modelling efforts, such as CMIP5 and CORDEX. Internally, each ESGF Node is composed of a set of services and applications for secure data/metadata publication and access and user management. Moreover, the search service offers an intuitive tool based on search categories or "facets" (e.g. "model" or "project"), through which users are enabled to constrain their search results.

For these features, ESGF has been selected as the most appropriate Information System for TESSA data dissemination and, above all, its visibility in the climate community.

### 4.1 Implementation of the Metadata Service

In this phase, the metadata profiles for the description of datasets produced by TESSA models have been formalized; moreover, the installation of a dedicated ESGF Data Node has been performed, customized on the requirements and specificities of TESSA project.

For its characteristics of simplicity and interoperability, the GeoNetwork Metadata Editor (geo) has been chosen as metadata editing tool for the description of datasets produced by TESSA models. In fact, GeoNetwork opensource is a multi-platform metadata catalogue application, designed to connect scientific communities and their data through the web environment. This software provides an easy to use web interface to search geospatial data across multiple catalogs, to publish geospatial data using the online metadata editing tools, to manage user and group accounts and to schedule metadata harvesting from other catalogs. In particular, GeoNetwork provides a powerful metadata editing tool that supports the ISO 19115 metadata profile and provides automatic metadata editing and management. Profiles can take the form of templates that is possible to fill in by using the Metadata Editor. Once the metadata profile has been compiled, it is also be possible to link data to related metadata description and to set the privileges to view metadata and to download the data.

As an example, Fig.4 reports an extract of the metadata profile compiled for the description of AFS simulation dataset.

Thus, this metadata profile represents a very powerful but, at the same time, user-friendly tool for the activities of search & discovery of the data produced in the frame of TESSA project. As stated before, the TESSA Information System also includes a dedicated ESGF Data Node, installed and properly configured for the project requirements. First of all, the web-interface has been customized by inserting the description of the project and other related information. However, the most important step has been the mapping of TESSA data properties onto the "search categories or facets", through which users can navigate the archive. In particular, new facets such as "spatial_resolution" and "spatial_domain" have been defined, in order to better fit the specificity of TESSA datasets. These new facets have been added to configuration files and so they are now available in the web-interface of TESSA Data Node to support data search. In parallel, specific global attributes have been inserted in all



output files of TESSA models, in order to provide the metadata values for the corresponding search facets. These features are schematically represented in Fig.5.

## 5   The CDAM: Complex Data Analysis Module

The CDAM component has been designed with the aim to optimize the DSS job submissions in a multi-channel (web or mobile) environment. This component exploits existing technologies, widely spread in client-server architectures, in order to simplify the complexity of the overall system, increasing the flexibility of the involved modules and the separation of concerns.

The Fig.6 shows an overview of the workflow related to TESSA decision support systems and the components involved in the operational chain.

The actors of such a scenario are (i) *the user*, interacting with the system using his mobile device or his computer, (ii) the *SSA (Situational Sea Awareness)* platform, and (iii) *the Complex Data Analysis Module* which exploits a cluster infrastructure for the job execution.

The user sends a request to the SSA platform by using a desktop or mobile application related to a specific DSS. The request contains a set of parameters that will be parsed by the SSA platform and sent to the CDAM through a secure communication channel. Once received the request, the CDAM Gateway is responsible for preparing the environment for the job submission and for sending the parameter to the underlying cluster infrastructure which executes the DSS algorithm and, at the end, sends a message to the SSA platform with the result of the processing.

### 5.1   The CDAM Gateway

The CDAM Gateway is designated to be the entry point for the job submission. It is responsible for managing the incoming requests on the basis of the different DSSs and for interacting with the underlying cluster infrastructure for the algorithm execution. The CDAM Gateway module is constituted by two components, the Web Interface and the CDAM scheduler.

The Web Interface provides the SSA platform with a unique and uniform interface since every request is served in the same way. In order to secure the communication channel between the SSA Platform and the CDAM Gateway, two levels of security constraints have to be satisfied. The former concerns the authentication/authorization method based on the Basic Authentication mechanism; the sender of the request (in this case the SSA Platform) needs to authenticate itself by providing the proper username and password. Moreover, the transfer of such information is protected by the HTTP protocol with SSL encryption (HTTPS). The latter regards a specific OS firewall policy which enables incoming communications from well-known IP range.

The CDAM scheduler is composed of a series of modules, one for each DSS or service that has been implemented. In particular, three types of services has been defined: submission of a job related to a specific DSS, deleting of an active job and an echo service, to check if the job submission service is available and to evaluate the response time.

The main tasks of each DSS specific module are:

1. to check the parameters received from the Web Interface;



2. to perform the setup of the environment on the cluster infrastructure, by creating the directories required for the algorithm execution;

3. to contact the execution host with the correct parameters.

Both Web interface and CDAM Scheduler modules are based on a logging mechanism for keep track of all the operations
performed and the errors occurred during the process.

## 5.2  The CDAM Launcher

The CDAM launcher is hosted on the entry point of the cluster infrastructure and has the responsibility to properly manage the DSS submission on the cluster.

As the CDAM Scheduler, the launcher is composed of one module for each DSS or service and performs the following
operations: it checks the parameters received from the scheduler, it prepares the files that will be used as input for the algorithm execution, it performs the job submission, and finally, it sends the result of the processing to the SSA platform.

It is worth noting that the design and implementation of the CDAM stack is independent from the software modules designated to start the submission; indeed, it provides general interfaces to perform remote submissions on a cluster environment. In TESSA project such a task is performed by the SSA platform which manages the requests incoming from the users and
interacts with the CDAM for the DSS algorithms execution.

## 6  Operational Activity: TESSA Data Platform Use Cases

The TESSA data platform developed within the TESSA project represents an innovative solution not only as it provides an efficient access to the data but, above all, as it exploits and integrates different technologies to fully satisfy any potential user's need at operational level. In fact, the three main components of the platform (DAS, Metadata Service and CDAM)
are not isolated blocks but interact each other supporting, with a high efficiency level, a number of services, such as the operational chain of Sea Conditions (sea, a), the automatic publication of datasets on the data nodes ESGF and the Situational Sea Awareness DSS.

### 6.1  Automatic publishing procedure on the ESGF Data Node

An automatic procedure for data publishing on the ESGF Data Node has been designed and implemented in order to publish
updated products on the TESSA ESGF portal.

This procedure is based on the strict interaction between the DTS (Data Transformation Service) and the Metadata Service. As first step, DTS applies a pre-processing on the input data that are mainly output of the hydrodynamic sub-regional models and, specifically, daily simulations and forecasts for the Adriatic, Ionian and Tyrrhenian Seas with different spatial and temporal resolution. This step is fundamental to make data compliant for the publication on the ESGF Data Node and involves
the insertion of specific global attributes to provide the metadata values for the corresponding search facets. Once the pre-





processing of the input data has been performed, the DTS modules alert the DAS component to publish the involved datasets on the ESGF node.

Fig.7 shows the mentioned procedure in terms of a workflow of operational tasks: the delivery of TESSA outputs (simulations and forecasts, hourly and daily) from the production host to the target host, the automatic pre-processing of datasets by DTS operations and the upload of the processed data to the target directory.

On the Metadata Service side, a controller module performs the publication of the different types of data preprocessed by DTS; specifically, the controller is a daemonized module that once a day sets up the directories *mapfiles/bulletindate* and runs in background the procedures for the publication of the outputs of the models.

## 6.2 DTS automatic procedures for DSSs and numerical models transformations

The DTS is also responsible for pre-processing the input data for the different DSSs, a set of tools for supporting users in making decisions or managing emergency situations and suggesting a strategy of intervention (namely Witoil(Dominicis et al., 2013b), VISIR(Mannarini et al., 2016), (Dominicis et al., 2013a), EarlyWarning). Specifically WITOIL predicts the transport and transformation of real or hypothetical oil spills in the Mediterranean Sea, VISIR suggests the best nautical routes in any weather-marine condition, and Early Warning manages alert in case of extreme events by providing a daily update on weather (wind, air temperature) and oceanographic conditions (wave height, sea level, water temperature). In detail, the raw data, downloaded using the DAS routines as described in Sect. 3, are supplied to the automated pre-processing chains of the DTS, responsible to apply a series of transformations on the data in order to suit the specific DSS input requirements. At the end of the transformation process, the system is able to manage the different on-demand DSS requests starting the related algorithms.

As an example, Witoil requires files of Temperature, Currents and Wind at sea surface; in particular, the marine raw data are subjected to a spatial subsampling selecting only the first 10 levels of depth. For the computation of the least-time nautical route, performed by VISIR, every day a single dataset of 10-days forecasts is provided; in this particular case, the DTS concatenates significant wave height, peak wave period, and wave direction fields. For EarlyWarning, the pre-processing procedure applied to the files is very simple and consists on a data unzipping to create a data block useful to the execution of the algorithm. The DTS is responsible to apply all the mentioned processing on the input data in order to let the different DSS operational chains faster and more efficient exploiting the automation provided by the DAS. Fig.8 and 9 show a nautical route computed by VISIR and a sea level forecast for the city of Venice computed by the EarlyWarning DSS.

The DAS is also extensively used by the DTS automatic processes for activating the sub-regional models production (i.e. AFS, AIFS, and SANIFS). The DAS modules are responsible for periodically downloading the source raw data from the external data provider. Once input data are available, the operational forecasting system is activated by specific DTS modules in order to schedule and manage the sub-regional models execution and to properly synchronize the pre and post-processing operational chains.

The interaction between the DAS and DTS is also important for the automated production of maps created starting from the outputs of the sub-regional model AFS, AIFS and SANIFS by means of dedicated operational chains.



### 6.3  Sea-Conditions operational chain

Concerning the SeaConditions operational chain, the main aim of the DTS is to create the properly processed and formatted data to visualize 5-days of weather-oceanographic forecasts with 3h of temporal resolution for the first two days and half and 6h for the others.

Dedicated DAS modules, day by day, download the source raw data from the external providers, store them into the data archive and provide them as input to the Sea-Conditions operational chains managed by DTS; post-processed data are then moved to an FTP Server. SSA Platform procedures are responsible to download them as input for the creation of the maps published on the Sea-Conditions web-portal.

This operational chain is based on a workflow manager implemented by a daemonized routine called DTS_SCHEDULER.
This daemon manages the parallelization of the data processing tasks using stage-in and stage-out folders as interchange layers with the data archive. The different DTS production phases are represented in Fig.10, including:

- Pre Elaboration Phase: involved in the data splitting by time and by variables;

- Statistic Phase: the aim is to compute some statistics as minimum, maximum, mean and standard deviation;

- Interpolation Phase: the split data resulting from the first phase are filtered by time and some of them are interpolated in
order to increase their spatial resolution;

- Compression Phase: the data resulting from the Interpolation Phase are compressed: Lempel-Ziv deflation, NetCDF4 packing algorithm (from the typical NC FLOAT to NC SHORT reducing the size by about 50%), file zipping with gzip are operations that generally lead to a 70-90% reduction in file size. If original NetCDF file occupies 1GB, at the end of packing and zipping process it will occupy approximately 100MB.

## 7  Conclusions

One of the main objectives of the TESSA project was to develop a set of operational oceanographic services for the Situational Sea Awareness. To reach this goal, advanced technological and software solutions are the backbone for the real time dissemination of the oceanographic data to a wide range of users in the maritime sector, tourism, and scientific community. In this framework, the relevance of the TESSA Data Platform is apparent, through its capacity to provide efficient and secure data
access and strong support to high-level services, such as the DSSs.

In general, the main components of this platform have been designed and developed by taking into account and satisfying a large number of requirements such as: transparency, robustness, efficiency, fault tolerance, security, data delivery and browsing and, above all, interoperability. Moreover, it is important to emphasize that this platform represents a unique and innovative example of how different components interact with each other to support operational services for the Situational Sea Awareness,
such as "Sea-Conditions" and the other DSSs (VISIR, WITOIL, Ocean-SAR, EarlyWarning).



Therefore, these features make the TESSA Data Platform a valid prototype easily adopted to provide an efficient dissemination of maritime data and a consolidation of the management of operational oceanographic activities.

*Acknowledgements.* This research was funded by the project TESSA (Technologies for the Situational Sea Awareness; PON01_02823/2) funded by the Italian Ministry for Education and Research (MIUR).



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

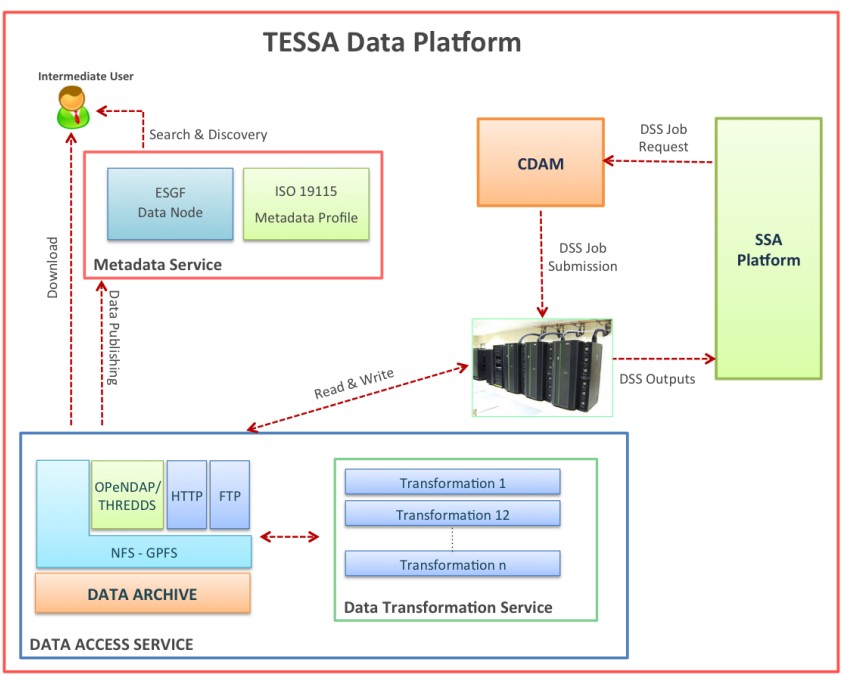

**Figure 1.** The TESSA Data Platform.



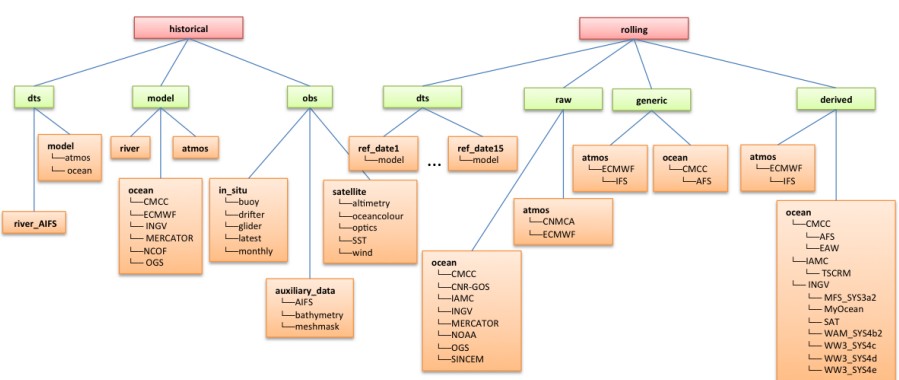

**Figure 2.** High-level representation of the Data Archive.

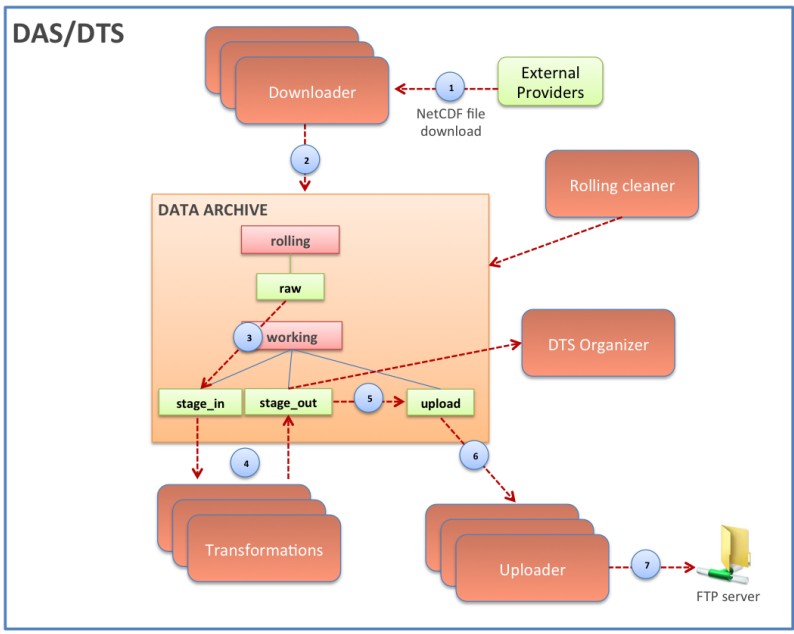

**Figure 3.** Interactions between the DAS, the DTS and the Data Archive.



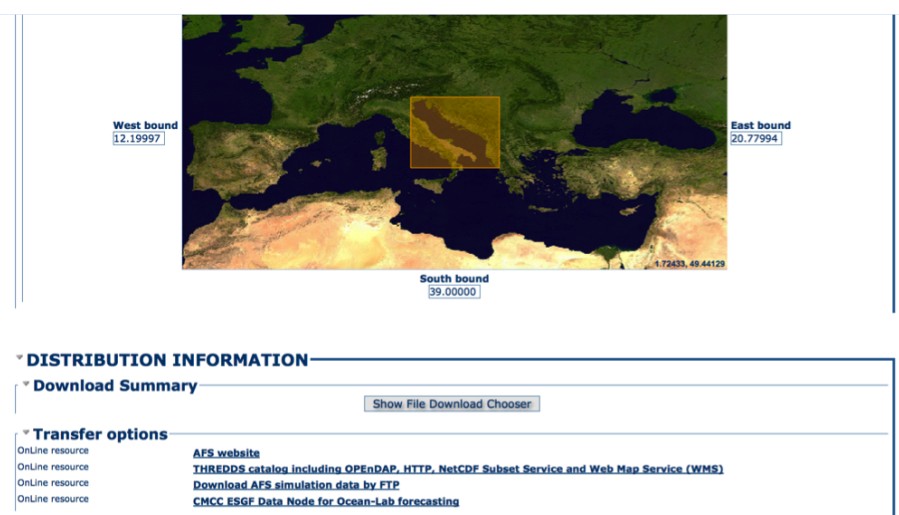

**Figure 4.** Metadata profile for an example of AFS dataset.

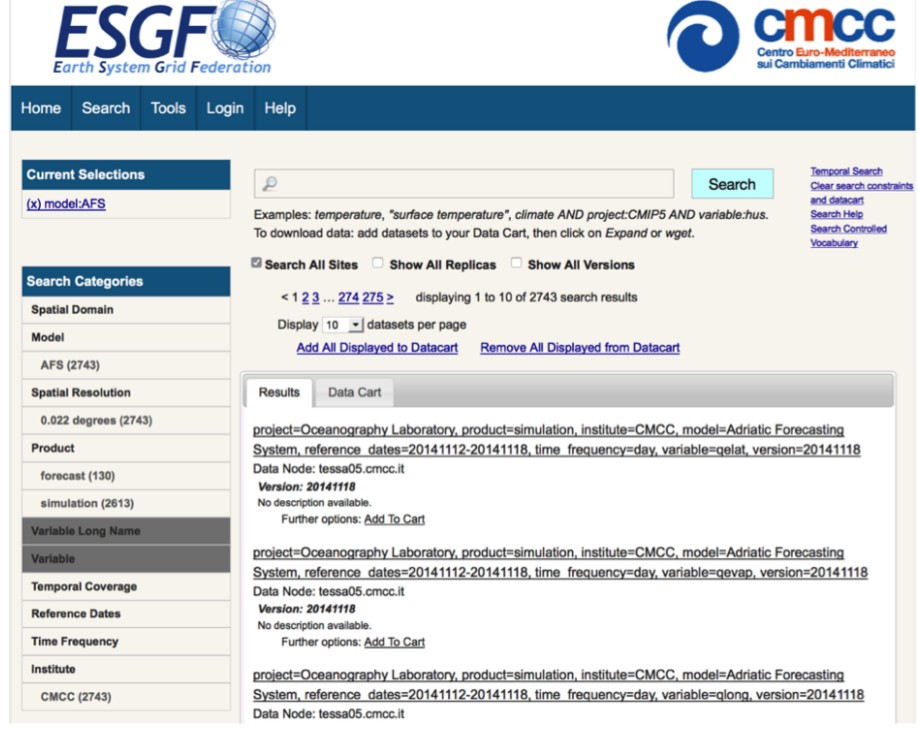

**Figure 5.** List of facets for AFS products as published in the ESGF Data Node for TESSA.





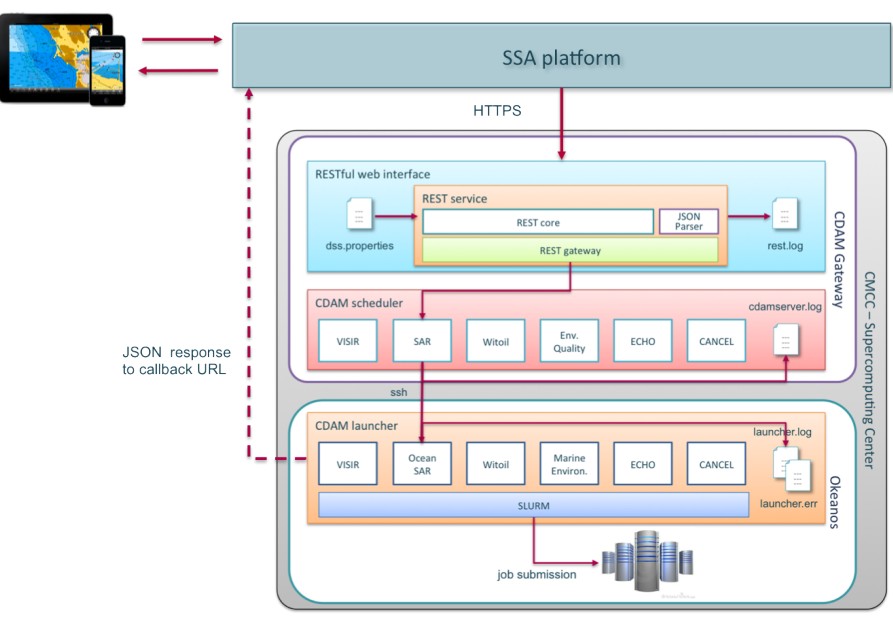

**Figure 6.** DSS request submission and interactions with the CDAM.

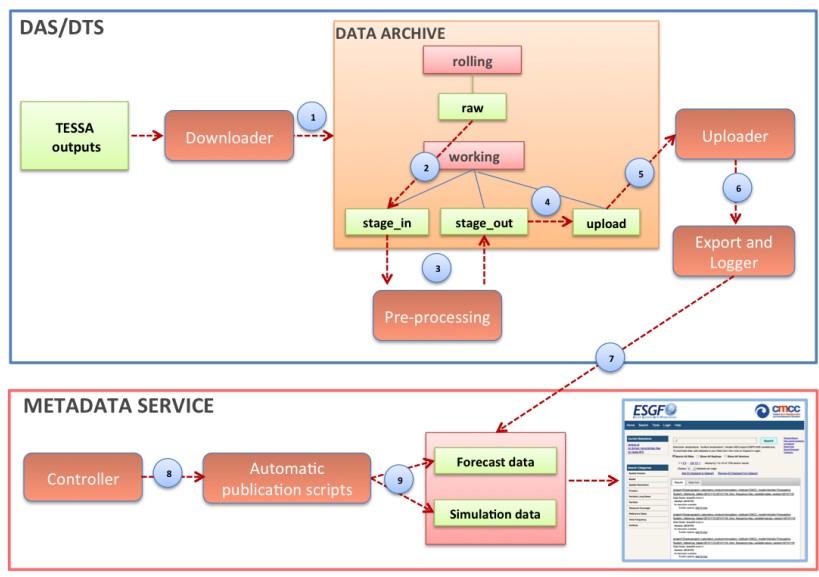

**Figure 7.** Operational chain for ESGF data publication.



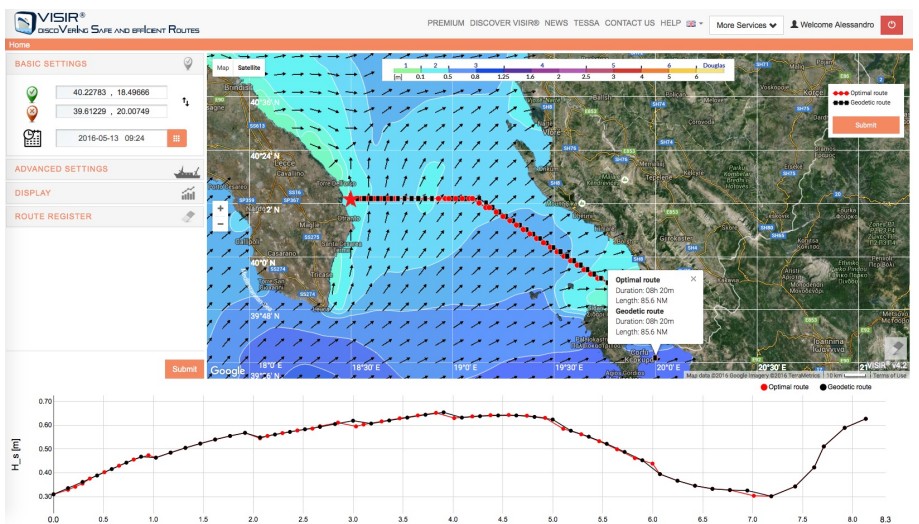

**Figure 8.** A motorboat route computed by VISIR.

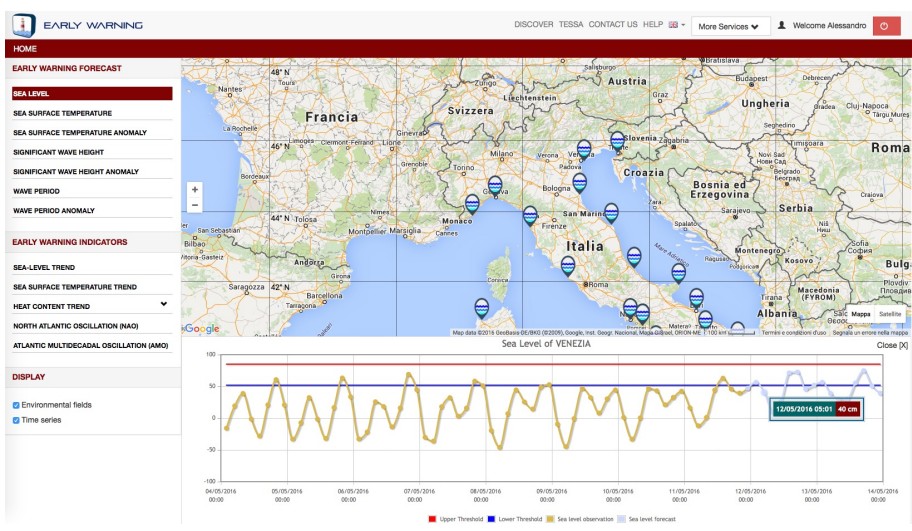

**Figure 9.** EarlyWarning forecast for the sea level of Venice.





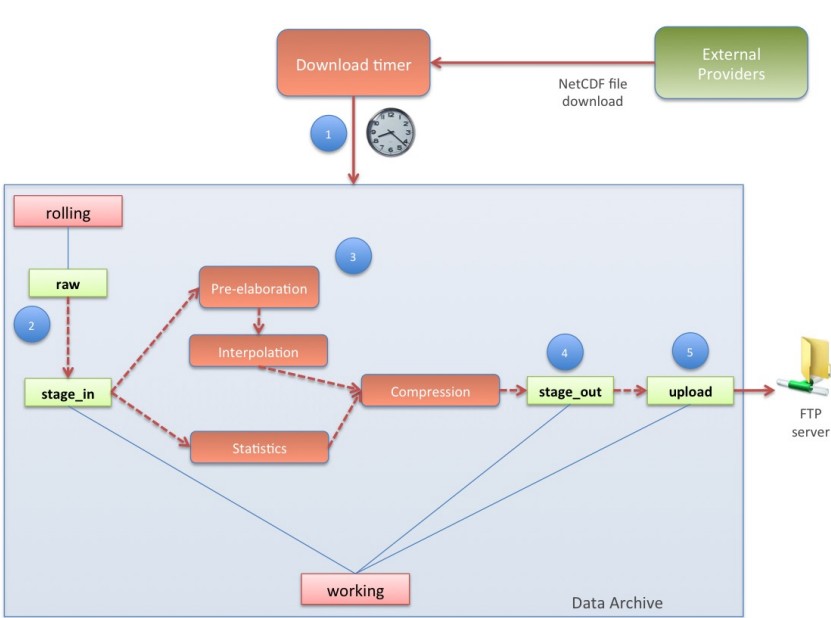

**Figure 10.** Operational chain for Sea-Conditions data preparation.