# Peer review of "A multi-service data management platform for scientific oceanographic products"

_Natural Hazards and Earth System Sciences, 2016_

## Short Comment (SC1) · 4 Aug 2016

Dear Authors,

I appreciate very much your involving in oil spill modeling, MEDSLIK-II, and WITOIL. I've found it both professional and very promising. Would you please fix a name of Michela De Dominicis in the "References" Section of your manuscript?

---

## Referee Comment (RC1) · Anonymous Referee #1 · 17 Sep 2016

The paper describes the data management platform developed by the TESSA project to provide a near real-time access to oceanographic data that enable their usage on Situational Sea Awareness services.

The presented solution adopted already existing systems, IT solutions and standards creating an innovative system that allows offering a wide set of services/features to the users in an integrated environment enabling a new way to make science.

Below some comments:

1) Nowadays, each data management system should demonstrate how it satisfies the FAIR Guiding Principles for scientific data management. The following paper is considered as a sort of reference for these principles: http://www.nature.com/articles/sdata201618. It would be an added value for the paper describing how the TESSA platform respects the guidelines described there.

2) In the introduction, authors refer to three standards: ISO 19115, ISO 19139, IN-SPIRE. In the rest of the paper is described how ISO 19115 is used but it is not clear how the other two standards have been implemented.

3) There are no references for these three standards in the introduction although they are listed in the reference list at the end (unless INSPIRE).

4) There is no related work sections. Maybe something could be added in the introduction.

5) The paragraph in pag. 3, lines 25-29 should be splitted in 2 paragraphs, it is not easy to read: "In particular, the THREDDS includes the WMS - Web Map Service (wms, 2006), for the dynamic generation of maps starting from geographically referenced data, the direct downloading using the HTTP protocol, and the OPeNDAP (ope) service, which is a framework that aims to simplify the sharing of scientific information on the web by making available local data from remote connections, providing also variables selection and spatial and temporal subsetting capabilities."

6) Reference list should be enriched.

7) Typo – Pag 1, line 20: "which is is strategically" to "which is strategically"

8) Typo – Pag 3, line 23: "a set of services that allow" to "a set of services that allows"

9) Typo – Pag. 9, line 11: "a strategy of intervention ( namely" to "a strategy of intervention, namely"

10) Typo: Pag. 9, line 11 and 19: "Witoil" to "WITOIL"

11) Typo: Pag. 9, line 19: "files of Temperature, Currents and Wind at. . ." to "files of temperature, currents and wind at. . ."

---

## Referee Comment (RC2) · Anonymous Referee #2 · 19 Sep 2016

The paper describes the standard based data management platform developed within framework of the TESSA project that targets scientific users and the Situational Sea Awareness high-level services.

Please find some comments below:

The related work subsection is missing.

Having in the introduction more information on the challenges/problems to be solved, would help in understanding the choices done.

In architecture subsection authors refer to "large number of requirements such as: transparency, robustness, efficiency, fault tolerance, security" , however it is not clear how transparency, robustness, fault tolerance is tackled in the proposed solution. (still

is being mentioned in conclusions)

Please rephrase/divide following sentences (for better readability):

" In particular, the THREDDS includes the WMS - Web Map Service (wms, 2006), for the dynamic generation of maps starting from geographically referenced data, the direct downloading using the HTTP protocol, and the OPeNDAP (ope) service, which is a framework that aims to simplify the sharing of scientific information on the web by making available local data from remote connections, providing also variables selection and spatial and temporal subsetting capabilities"

"Dedicated DAS modules, day by day, download the source raw data from the external providers, store them into the data archive and provide them as input to the Sea-Conditions operational chains managed by DTS; post-processed data are then moved to an FTP Server. "

Also the number of abbreviations/technologies mentioned in the abstract makes it a bit hard to read.

---

## Author Comment (AC1) · 31 Oct 2016

Dear Svitlana, thank you for your comment and appreciation. As you suggest, we will fix the name of Michela De Dominicis in the "References" section: De Dominicis, M.

---

## Author Comment (AC2) · 31 Oct 2016

Referee general comment: The paper describes the data management platform developed by the TESSA project to provide a near real-time access to oceanographic data that enable their usage on Situational Sea Awareness services. The presented solution adopted already existing systems, IT solutions and standards creating an innovative system that allows offering a wide set of services/features to the users in an integrated environment enabling a new way to make science.

Authors' Response: Our thanks to the referee, which his/her valuable suggestions improved the quality of the paper. Our replies are as follows.

1) Nowadays, each data management system should demonstrate how it satisfies the FAIR Guiding Principles for scientific data management. The fol-

[Figure]

lowing paper is considered as a sort of reference for these principles: http://www.nature.com/articles/sdata201618. It would be an added value for the paper describing how the TESSA platform respects the guidelines described there.

- Authors' response:

Following the suggestion of the referee we will add this text in the conclusion:

The FAIR Guiding Principles for scientific data management and stewardship [FAIR] faces the lack of a shared, comprehensive, broadly applicable, as well as articulated guideline concerning the publication of scientific data. Specifically, it identifies in Findability, Accessibility, Interoperability and Reusabiliy the four principles which need to be addressed by data (and associated metadata) to be considered as a good publication. The designed Data Platform addresses many of these requirements; first of all, it relies on THREDDS for the publication of the TESSA final products (Findable and Accessible), while the OPeNDAP service exposes a standard DAP interface (Interoperable). In addition, the ISO19115 standard has been adopted for the definition of a metadata profile containing the mandatory elements of the "core metadata set for geographic datasets" (Interoperable and Reusable) and the search & discovery of data are supported by the ESGF Data Node and by the Geonetwork Information System (Findable and Accessible). Finally, access to the DSS methods is provided by a machine readable interface exploiting the JSON format (Accessible).

Reference:

FAIR: http://www.nature.com/articles/sdata201618

2) In the introduction, authors refer to three standards: ISO 19115, ISO 19139, INSPIRE. In the rest of the paper is described how ISO 19115 is used but it is not clear how the other two standards have been implemented.

- Authors' response:

The text (page 2 row 8) refers to the project MyOcean as a platform that takes into
account different standards (ISO 19115, ISO 19139 and INSPIRE) while the Metadata profile adopted for the TESSA project relies on the ISO 19115 standard for its definition and the ISO 19139 standard for the related xml representation (page 5 row 19). In accordance with the referee suggestion, we will move the MyOcean description in the Related Work section (see point 4). For better explaining how ISO 19139 is implemented we will modify the text (page 6 rows 21-24) as follows:

In particular, GeoNetwork provides a powerful metadata editing tool that supports the ISO 19115 metadata profile and provides automatic metadata editing and management. Profiles can take the form of templates that it's possible to fill by using the Metadata Editor. Once the metadata profile has been compiled, it is also possible to generate the related XML schema compliant with ISO 19139, to link data to related metadata description and to set the privileges to view metadata and to download the data.

3) There are no references for these three standards in the introduction although they are listed in the reference list at the end (unless INSPIRE).

- Authors' response:

References on ISO standards are in the Chapter 4 (page 5, rows 6 and 19). Following your suggestion we will move that references in the Related Work section. In addition, we will add the INSPIRE references in the reference section.

1. INSPIRE web-site - http://inspire.ec.europa.eu/

2. INSPIRE directive -

http://eur-lex.europa.eu/JOHtml.do?uri=OJ:L:2007:108:SOM:EN:HTML

4) There is no related work sections. Maybe something could be added in the introduction.

- Authors' response:

A section containing the related work will be added to the document.

Related work

The need for marine and oceanic data management supporting the Situational Sea Awareness and the operational oceanography has led to the definition and development of various platforms that provide different types of data and services. In this context, the MyOcean project (MyO)(MyO2) can be mentioned, as this project implementation was in line with the best practices of the GMES/Copernicus framework (cop). Specifically, regarding data access, MyOcean provides the scientific community with a unified interface designed to take into account various international standards (ISO 19115 (iso, 2003a), ISO 19139 (iso, 2003b), INSPIRE (inspire1)(inspire2)). Concerning the data management, MyOcean relies on OPeNDAP/THREDDS for tasks like map subsetting and FTP for direct download.

In addition, a valid solution is represented by the Earth System Grid Federation (ESGF) (Cinquini et al., 2014)(esgf), a federated system used as metadata service with advanced features, that will be described later. EMODNET MEDSEA Checkpoint (emodnet) (Moussat et al.) is another solution supporting data collection and data search & discovery: it exploits a checkpoint browser and a checkpoint dashboard, which presents indicators automatically produced from information database. SeaDataNet [sea, b] represents a distributed Marine Data Management Infrastructure able to manage different large datasets related to in situ and remote observation of the seas and oceans. Through a distributed network approach, it provides an integrated overview and access to datasets provided by 90 national oceanographic and marine data centers. Finally, another efficient solution has been developed within the project CLIPC [clipc]. It provides access to climate information including data from satellite and in-situ observations, transformed data products and climate change impact indicators. Moreover, users can exploit the CLIPC toolbox to generate, compare, manipulate and combine indicators, create an user basket or launch new jobs for indices calculation.

These systems support a wide range of functionalities such as data access, classification, search & discovery and download by means of a distributed architecture. Graphs and maps production are also supported and some of these propose tools to manipulate and combine datasets.

The proposed architecture of the TESSA Data Platform should offer superior and fast data management functionalities, producing datasets suitable for serving different kinds of services in near real-time: access, search & discovery are just some of the features offered. The developed system, in fact, provides a common interface for submitting complex algorithms in an hpc environment using standard approaches and formats exploiting a unified infrastructure for supporting different scenarios and applications.

Related work references:

MyO: MyOcean, http://marine.copernicus.eu/

MyO2: P. Bahurel, F. Adragna, M. J. Bell, F. Jacq, J. A. Johannessen, P. Le Traon, N. Pinardi, J. She: Ocean Monitoring and forecasting core services, the European MyOcean example, Proceedings of OceanObs'09, 2009

cop: Copernicus, http://www.copernicus.eu/.

iso, 2003a: ISO 19115:2003 Geographic information - Metadata, Tech. rep., International Organization for Standardization (TC 211), 2003a.

iso, 2003b: ISO 19139:2007 Geographic information - Metadata - XML Schema Implementation, Tech. rep., International Organization for Standardization (TC 211), 2003b.

inspire1: INSPIRE web-site - http://inspire.ec.europa.eu/

inspire2: INSPIRE directive -

http://eur-lex.europa.eu/JOHtml.do?uri=OJ:L:2007:108:SOM:EN:HTML

Cinquini et al., 2014: Cinquini, L., Crichton, D., Mattmann, C., Harney, J., Shipman, G.,

Wang, F., Ananthakrishnan, R., Miller, N., Denvil, S., Morgan, M., Pobre, Z. Bell, G. M., Doutriaux, C., Drach, R., Williams, D., Kershaw, P., Pascoe, S., Gonzalez, E., Fiore, S., and Schweitzer, R.: The Earth System Grid Federation: An open infrastructure for access to distributed geospatial data, Future Generation Computer Systems, ISSN 0167-739X, http://dx.doi.org/10.1016/j.future.2013.07.002, 2014

esgf: http://esgf.llnl.gov/

emodnet: http://www.emodnet-mediterranean.eu/

Moussat et al: E. Moussat, N. Pinardi, G. Manzella, F. Blanc: EMODnet MedSea Checkpoint for sustainable Blue Growth, EGU General Assembly, 2016

sea, b: SeaDataNet, http://www.seadatanet.org/

clipc: http://www.clipc.eu/home

Moreover, the introduction will be modified as follows (page 2 rows 3-14):

In a "data centric" perspective, in which different services, applications or users make use of the outputs of regional or global numerical models, the TESSA Data platform meets the request of near real-time access to heterogeneous data with different accuracy, resolution or degrees of aggregation. Specifically, the design phase has been driven by multiple needs that the developed solution had to satisfy. First of all, the need for a service that provides information about sea conditions 24 hours for 7 days a week at high and very high spatial and temporal resolution has been addressed by exploiting high performance and high availability hardware and software solutions.

In addition, the developed platform must to be able to support the requests of intermediate and common users. To this end, data must be available in the native and standard format (NetCDF) [netcdf1][netcdf2] as output of the oceanographic model, through a simple and intuitive platform but also suitable for machine-based interactions, in order to feed user-friendly services for displaying clear maps and graphs. At the end, the system has to provide on-demand services to support decisions; the users must be able

to interact with the datasets produced by the models in near-real time, so the platform has to provide services and datasets suitable for on-demand processing minimizing the downloading time and the related input file size.

References

netcdf1: http://www.opengeospatial.org/standards/netcdf

netcdf2: http://www.unidata.ucar.edu/software/netcdf/

5) The paragraph in pag. 3, lines 25-29 should be splitted in 2 paragraphs, it is not easy to read: "In particular, the THREDDS includes the WMS - Web Map Service(wms, 2006), for the dynamic generation of maps starting from geographically referenced data, the direct downloading using the HTTP protocol, and the OPeNDAP (ope) service, which is a framework that aims to simplify the sharing of scientific information on the web by making available local data from remote connections, providing also variables selection and spatial and temporal subsetting capabilities."

- Authors' response:

We will modify the paragraph as follows:

In particular, the THREDDS includes the WMS - Web Map Service (wms, 2006), for the dynamic generation of maps starting from geographically referenced data and the direct downloading using the HTTP protocol. Futhermore, it includes the OPeNDAP (ope) service, which is a framework that aims to simplify the sharing of scientific information on the web by making available local data from remote connections, providing also variables selection and spatial and temporal subsetting capabilities.

6) Reference list should be enriched.

- Authors' response:

References will be enriched as follows:

In the introduction:

netcdf1: http://www.opengeospatial.org/standards/netcdf

netcdf2: http://www.unidata.ucar.edu/software/netcdf/

In the related work section:

MyO2: P. Bahurel, F. Adragna, M. J. Bell, F. Jacq, J. A. Johannessen, P. Le Traon, N. Pinardi, J. She: Ocean Monitoring and forecasting core services, the European MyOcean example, Proceedings of OceanObs'09, 2009

inspire1: INSPIRE web-site - http://inspire.ec.europa.eu/

inspire2: INSPIRE directive -

http://eur-lex.europa.eu/JOHtml.do?uri=OJ:L:2007:108:SOM:EN:HTML

esgf: http://esgf.llnl.gov/

emodnet: http://www.emodnet-mediterranean.eu/

Moussat et al: E. Moussat, N. Pinardi, G. Manzella, F. Blanc: EMODnet MedSea Checkpoint for sustainable Blue Growth, EGU General Assembly, 2016

clipc: http://www.clipc.eu/home

In the conclusions:

FAIR: http://www.nature.com/articles/sdata201618

7)-11) Typos

- Authors' response:

Thanks to the referee for the typos detection. We will modify the text accordingly.
* * *

---

## Author Comment (AC3) · 31 Oct 2016

Referee general comment: The paper describes the standard based data management platform developed within framework of the TESSA project that targets scientific users and the Situational Sea Awareness high-level services.

- Authors' response:

We thank the referee for comments and interesting suggestions. We address the reported comments as follows.

1) The related work subsection is missing.

- Authors' response:

A section containing the related work will be added to the document.

[Figure]

The need for marine and oceanic data management supporting the Situational Sea Awareness and the operational oceanography has led to the definition and development of various platforms that provide different types of data and services. In this context, the MyOcean project (MyO)(MyO2) can be mentioned, as this project implementation was in line with the best practices of the GMES/Copernicus framework (cop). Specifically, regarding data access, MyOcean provides the scientific community with a unified interface designed to take into account various international standards (ISO 19115 (iso, 2003a), ISO 19139 (iso, 2003b), INSPIRE (inspire1)(inspire2)). Concerning the data management, MyOcean relies on OPeNDAP/THREDDS for tasks like map subsetting and FTP for direct download.

In addition, a valid solution is represented by the Earth System Grid Federation (ESGF) (Cinquini et al., 2014)(esgf), a federated system used as metadata service with advanced features, that will be described later. EMODNET MEDSEA Checkpoint (emodnet) (Moussat et al.) is another solution supporting data collection and data search & discovery: it exploits a checkpoint browser and a checkpoint dashboard, which presents indicators automatically produced from information database. SeaDataNet [sea, b] represents a distributed Marine Data Management Infrastructure able to manage different large datasets related to in situ and remote observation of the seas and oceans. Through a distributed network approach, it provides an integrated overview and access to datasets provided by 90 national oceanographic and marine data centers. Finally, another efficient solution has been developed within the project CLIPC [clipc]. It provides access to climate information including data from satellite and in-situ observations, transformed data products and climate change impact indicators. Moreover, users can exploit the CLIPC toolbox to generate, compare, manipulate and combine indicators, create an user basket or launch new jobs for indices calculation.

These systems support a wide range of functionalities such as data access, classification, search & discovery and download by means of a distributed architecture. Graphs and maps production are also supported and some of these propose tools to manipulate and combine datasets.

The proposed architecture of the TESSA Data Platform should offer superior and fast data management functionalities, producing datasets suitable for serving different kinds of services in near real-time: access, search & discovery are just some of the features offered. The developed system, in fact, provides a common interface for submitting complex algorithms in an hpc environment using standard approaches and formats exploiting a unified infrastructure for supporting different scenarios and applications.

Related work references:

MyO: MyOcean, http://marine.copernicus.eu/

MyO2: P. Bahurel, F. Adragna, M. J. Bell, F. Jacq, J. A. Johannessen, P. Le Traon, N. Pinardi, J. She: Ocean Monitoring and forecasting core services, the European MyOcean example, Proceedings of OceanObs'09, 2009

cop: Copernicus, http://www.copernicus.eu/.

iso, 2003a: ISO 19115:2003 Geographic information - Metadata, Tech. rep., International Organization for Standardization (TC 211), 2003a.

iso, 2003b: ISO 19139:2007 Geographic information - Metadata - XML Schema Implementation, Tech. rep., International Organization for Standardization (TC 211), 2003b.

inspire1: INSPIRE web-site - http://inspire.ec.europa.eu/

inspire2: INSPIRE directive -

http://eur-lex.europa.eu/JOHtml.do?uri=OJ:L:2007:108:SOM:EN:HTML

Cinquini et al., 2014: Cinquini, L., Crichton, D., Mattmann, C., Harney, J., Shipman, G., Wang, F., Ananthakrishnan, R., Miller, N., Denvil, S., Morgan, M., Pobre, Z. Bell, G. M., Doutriaux, C., Drach, R., Williams, D., Kershaw, P., Pascoe, S., Gonzalez, E., Fiore, S., and Schweitzer, R.: The Earth System Grid Federation: An open infrastructure

for access to distributed geospatial data, Future Generation Computer Systems, ISSN 0167-739X, http://dx.doi.org/10.1016/j.future.2013.07.002, 2014

esgf: http://esgf.llnl.gov/

emodnet: http://www.emodnet-mediterranean.eu/

Moussat et al: E. Moussat, N. Pinardi, G. Manzella, F. Blanc: EMODnet MedSea Checkpoint for sustainable Blue Growth, EGU General Assembly, 2016

sea, b: SeaDataNet, http://www.seadatanet.org/

clipc: http://www.clipc.eu/home

2) Having in the introduction more information on the challenges/problems to be solved, would help in understanding the choices done.

- Authors' response:

We will modify the introduction (page 2 rows 3-14):

In a "data centric" perspective, in which different services, applications or users make use of the outputs of regional or global numerical models, the TESSA Data platform meets the request of near real-time access to heterogeneous data with different accuracy, resolution or degrees of aggregation. Specifically, the design phase has been driven by multiple needs that the developed solution had to satisfy. First of all, the need for a service that provides information about sea conditions 24 hours for 7 days a week at high and very high spatial and temporal resolution has been addressed by exploiting high performance and high availability hardware and software solutions.

In addition, the developed platform must to be able to support the requests of intermediate and common users. To this end, data must be available in the native and standard format (NetCDF) [netcdf1][netcdf2] as output of the oceanographic model, through a simple and intuitive platform but also suitable for machine-based interactions, in order to feed user-friendly services for displaying clear maps and graphs. At the end, the system has to provide on-demand services to support decisions; the users must be able to interact with the datasets produced by the models in near-real time, so the platform has to provide services and datasets suitable for on-demand processing minimizing the downloading time and the related input file size.

References

netcdf1: http://www.opengeospatial.org/standards/netcdf

netcdf2: http://www.unidata.ucar.edu/software/netcdf/

3) In architecture subsection authors refer to "large number of requirements such as: transparency, robustness, efficiency, fault tolerance, security" , however it is not clear how transparency, robustness, fault tolerance is tackled in the proposed solution. (still is being mentioned in conclusions).

- Authors' response:

To address the referee comment, the text will be modified as follows:

Page 3, row 29 to insert:

To improve performance in terms of load balancing and increase the fault tolerance of the system, a multiplexed configuration for the THREDDS installation based on two different hosts has been used.

Page 3, rows 29-31 to modify:

In addition, a sub-component, the Data Transformation Service (DTS), represents a middleware responsible to apply different data transformations in order to make data compliant to the different protocols offered by the DAS and therefore available to the data consumers (users or services) in a transparent way.

Page 4, row 2 to modify:

It has been designed and implemented in order to ease the data search & discovery

phase exploiting their classification while, from an infrastructure point of view, data are stored on a GPFS [gpfs] file system (in RAID 6 configuration) able to provide fast data access and high level of fault tolerance.

Page 5, row 2, to insert:

With the aim to increase the performance and the robustness of the system, all the mentioned components work in a multi-threaded environment: each thread is responsible to manage distinct sets of files. To cope with erroneous input, a MD5 checksum is also used. It is worth noting that the whole process is performed in a transparent way with respect to the services that act as consumers of the produced datasets in order to provide data always updated to the latest versions.

Page 8, row 12 to modify:

It is worth noting that the design and implementation of the CDAM stack is independent from the software modules designated to start the submission; indeed, it provides general interfaces exposing a remote submission service able to hiding the hpc resources for concurrent execution of several jobs.

References

gpfs: Schmuck, Frank; Roger Haskin (January 2002). "GPFS: A Shared-Disk File System for Large Computing Clusters". Proceedings of the FAST 2002 Conference on File and Storage Technologies. Monterey, California, USA

4) Please rephrase/divide following sentences (for better readability): "In particular, the THREDDS includes the WMS - Web Map Service (wms, 2006), for the dynamic generation of maps starting from geographically referenced data, the direct downloading using the HTTP protocol, and the OPeNDAP (ope) service, which is a framework that aims to simplify the sharing of scientific information on the web by making available local data from remote connections, providing also variables selection and spatial and temporal subsetting capabilities".

"Dedicated DAS modules, day by day, download the source raw data from the external providers, store them into the data archive and provide them as input to the Sea-Conditions operational chains managed by DTS; post-processed data are then moved to an FTP Server."

- Authors' response:

We will rephrase the sentences as follows:

"In particular, the THREDDS includes the WMS - Web Map Service (wms, 2006), for the dynamic generation of maps starting from geographically referenced data and the direct downloading using the HTTP protocol. Futhermore, it includes the OPeNDAP (ope) service, which is a framework that aims to simplify the sharing of scientific information on the web by making available local data from remote connections, providing also variables selection and spatial and temporal subsetting capabilities."

"Dedicated DAS modules, day by day, download the source raw data from the external providers storing them into the data archive. Raw data are then provided to the Sea-Conditions operational chains managed by DTS; at the end post-processed outputs are moved to an FTP Server."

5) Also the number of abbreviations/technologies mentioned in the abstract makes it a bit hard to read.

- Authors' response

The abstract will be modified as follows:

An efficient, secure, and interoperable data platform solution has been developed in the TESSA project to provide fast navigation and access to the data stored in the data archive, as well as a standard-based metadata management support. The platform mainly targets scientific users and the Situational Sea Awareness high-level services such as the Decision Support Systems (DSS). These datasets are accessible through the following three main components: the Data Access Service (DAS), the Metadata

Service, and the Complex Data Analysis Module (CDAM). The DAS allows access to data stored into the archive by providing interfaces for different protocols and services for downloading, variables selection, data subsetting or map generation. Metadata Service is the heart of the information system of the TESSA products and completes the overall infrastructure for data and metadata management. This component enables data search & discovery, and addresses interoperability by exploiting widely adopted standards for geospatial data. Finally, the CDAM represents the back-end of the TESSA DSS by performing on-demand complex data analysis tasks.